# Real-Time Ultrasonography and the Evaluation of Static Images Yield Different Results in the Assessment of EU-TIRADS Categories

**DOI:** 10.3390/jcm12185809

**Published:** 2023-09-06

**Authors:** Dorota Słowińska-Klencka, Bożena Popowicz, Mariusz Klencki

**Affiliations:** Department of Morphometry of Endocrine Glands, Medical University of Lodz, Pomorska Street 251, 92-213 Lodz, Poland; bozena.popowicz@umed.lodz.pl (B.P.); marklen@tyreo.umed.lodz.pl (M.K.)

**Keywords:** thyroid, EU-TIRADS, real-time ultrasonography, static ultrasonography, thyroid cancer

## Abstract

The studies on the effectiveness of various TIRADS in the diagnostics of thyroid nodules differ in the method of ultrasound image assessment: real time (rtUS) vs. static ultrasonography (stUS). The aim of the study was to evaluate the impact of those two methods on the categorization of nodules in EU-TIRADS. Three experienced raters assessed 842 nodules in routine rtUS and reassessed with the use of sUS. Reproducibility of the assessment of malignancy risk features and categorization of nodules with EU-TIRADS was estimated with Krippendorff’s alpha coefficient (Kα). The reproducibility of EU-TIRADS categories on sUS in relation to rtUS was in range 70.9–76.5% for all raters (Kα: 0.60–0.68) with the highest reproducibility for category 3 (80.0–86.5%) and the lowest for category 5 (48.7–77.8%). There was a total disagreement of the identification of microcalcifications on sUS in relation to rtUS, a strongly variable reproducibility of marked hypoechogenicity (12.5–84.6%, Kα: 0.14–0.48) and a tendency toward more frequent identification of the non-oval shape on sUS. The percentage of agreement for each pair of raters in assigning the EU-TIRADS category on sUS was in the range 71.6–72.3% (Kα: 0.60–0.62). The method of sonographic image evaluation influences the nodule’s feature assessment and, eventually, the categorization within EU-TIRADS.

## 1. Introduction

Ultrasound imaging (US) of the thyroid is a standard procedure in the evaluation of thyroid nodules. It not only reveals nodules and precisely determines their size but also provides a means to assess some features of nodules that are more or less indicative of their malignancy. These data aid the optimal selection of nodules for the fine-needle aspiration biopsy (FNA) [1,2]. US is also helpful when the cytological outcome is indeterminate, as it improves the estimation of the malignancy risk of such a nodule [3,4]. Currently, there are some systems in use for the assessment and better stratification of malignancy risk based on the nodule’s features identified on US. These systems are usually called TIRADS, which is an acronym denoting the Thyroid Imaging Reporting and Data System. All the systems employ joint analysis of several sonographic features of the nodule, because when considered separately, these features do not have suitable both sensitivity and specificity at the same time. The system commonly employed in Europe is called EU-TIRADS, and it is recommended by the European Thyroid Association (ETA) [1]. The EU-TIRADS uses a 5-grade scale, in which the high specificity US malignancy features include marked hypoechogenicity, irregular shape, irregular margins and microcalcifications. Similar systems were concomitantly developed in Asia and the U.S. The most popular system in Asia is K-TIRADS, which is recommended by the Korean Society of Thyroid Radiology [5]. The two most popular American systems are the system endorsed by American Thyroid Association and the system recommended by American College of Radiology [2,6]. The wide spread of TIRADSs stimulated publication of numerous reports that compared their effectiveness on the same material, as well as among different centers [7,8,9,10,11,12,13]. Their conclusions are often contradictory, especially in the case of nodules with indeterminate cytology [3,4,14,15,16]. Upon detailed analysis of these reports, one may conclude that these discrepancies result from both external factors, like geographical variation in the epidemiology of thyroid nodules, as well as internal, modifiable factors. Among the latter, differences in assessment of sonographic images are most notable. In some centers, the assessment is performed in real time (real time ultrasonography) [7,9,13,14,17], while in others, cine loops or stored images are used for that purpose (static ultrasonography) [3,4,10,11,12,15,16]. Unfortunately, reports from such studies frequently do not include precise information on the number of stored images, used sections (transverse, longitudinal) or uniform imaging settings if applied [3,4,10,11]. Thus, we decided to evaluate the impact of those two different methods of the assessment on categorization of nodules in EU-TIRADS, which is routinely used in our department. The main aim of the study was to compare results of nodule classification into particular EU-TIRADS categories in relation to the method of sonographic image evaluation: real time vs. static ultrasonography. An additional aim was to evaluate reproducibility of the classification based on static ultrasonography.

## 2. Materials and Methods

### 2.1. Data Collection

All the examinations were made in a single center with over 20 years of experience in US and FNA of the thyroid. In our routine practice, each FNA is preceded by a thorough US examination when the nodule’s presentation and its relevant sonographic features are evaluated and then the nodule is classified into an appropriate EU-TIRADS category [1]. Particular features are identified according to a unified pattern that has been used at our department for many years: (1) echogenicity categorized as marked hypoechogenicity (darker than surrounding strap muscles), hypoechogenicity (as compared to the normal thyroid) or other (the lowest echogenicity area, irrespective of its volume share, is considered); (2) composition categorized as solid (>90% solid), solid–cystic (more solid than cystic), cystic–solid (more cystic than solid), spongiform, or cystic composition involving the entire nodule; (3) shape categorized as non-oval (taller than wide or round) or oval; (4) irregular margins (including microlobulated or spiculated) or other; (5) calcifications categorized as microcalcifications (around 1 mm in size without posterior shadowing, located in the solid component of the nodule), macrocalcifications or rim calcifications. We use a custom computer program dedicated to collecting detailed information on examined nodules in a database (like their location, echogenicity, dimensions, etc.). This software is integrated with a DICOM server on which at least one still image of the biopsied nodule is stored that documents the presence of key features of the nodule. All images are taken at the same preset optimized for the thyroid imaging. In most cases, two images are obtained: one at the transverse section and the other at the longitudinal section. Less often, only a single image at the transverse section is stored.

The study included a subset of data collected in the way described above, between October 2021 and September 2022, by three doctors with at least 15 years’ experience in performing US and FNA of the thyroid gland (they were denoted as the raters A, B and C). Nodules in patients with previous surgical or radioiodine thyroid treatment or positive neck irradiation history were excluded. At that stage, the study included 876 nodules. The same three raters reassessed all those nodules with the use of previously stored images in December 2022. The presence of the nodule’s features considered in EU-TIRADS was evaluated and the nodules were assigned to appropriate categories of the system. A dedicated application module had been created for that purpose (by one of us—MK), which facilitated presentation of images and collection of results of the assessment. The raters were not aware of the originally assigned EU-TIRADS category nor FNA outcome while evaluating images. There were 34 nodules excluded from further analysis because at least one investigator found it impossible to assess all relevant sonographic features in the image (usually due to a large size of the nodule or the presence of extensive rim calcifications or poor quality of the image). Eventually, the analysis included 842 nodules revealed in 641 patients (Table 1). The numbers of nodules originally evaluated in real-time sonography by each rater were similar (A: 268 nodules, B: 282; C: 292). In the examined set of nodules, there were 151 nodules (17.9%) with all three diameters ≤ 10 mm. There were two images available for the evaluation in the case of 545 nodules, one at each section (longitudinal and transverse) and only one image (transverse section) in the case of 297 nodules.

All US was performed with the use of the Aplio α ultrasound system (Canon Medical Systems Europe B.V., Zoetermeer, The Netherlands) with a matrix linear probe 14L5. The selection of nodules for FNA was performed according to ETA guidelines—EU-TIRADS, but in some cases, preferences of the patient or the referring endocrinologist were followed and thyroid nodules smaller than indicated, in particular, EU-TIRADS categories, were biopsied (with a diameter of at least 5 mm). In all cases, two aspirations of a nodule were done and the smears were fixed with 95% ethanol solution and stained with hematoxylin and eosin. The results of FNA were formulated according to the Bethesda classification [18,19].

### 2.2. Reproducibility of the Assessment on Static US in Relation to Real-Time US

The percentage of concordant assignments of EU-TIRADS category on static US (sUS) was determined in relation to real-time US (rtUS). The reproducibility was further assessed in more detail, considering the nodule’s size (≤10 mm vs. larger) and number of images (transverse section vs. both sections). The performance of each rater was analyzed separately on the subset of nodules evaluated on rtUS by that person. The number and percentage of true positive classifications into each EU-TIRADS category and true positive identification of examined sonographic features were determined for each rater on sUS in relation to the outcome of standard rtUS. Then, possible causes of disagreement between assigned EU-TIRADS categories were analyzed by the comparison of frequencies of EU-TIRADS categories as indicated by each rater, as well as frequencies of particular sonographic features on rtUS and sUS. The following features were evaluated: (a) four high-risk features: marked hypoechogenicity, non-oval shape, irregular border and microcalcifications, (b) three other features relevant for EU-TIRADS: hypoechogenicity, solid echostructure and cystic/spongiform echostructure, (c) macrocalcifications, as determined by particular raters. The non-oval shape feature in sUS was determined by the general impression of the rater and not by exact measurements recorded during rtUS. 

Finally, the incidence of indeterminate or malignant categories of FNA diagnosis (categories III–VI in the Bethesda classification) was determined for each EU-TIRADS category in relation to the method of sonographic features assessment (rtUS vs. sUS).

### 2.3. Reproducibility of Nodule Assessment on Static US

The concordance of the classification of sUS images into EU-TIRADS categories was evaluated between the raters. The percentage of cases with complete agreement of categorization among all three raters was determined. Similar analyses were performed for three possible pairs of participating raters. The concordance of categorization was also analyzed in relation to the nodule size (≤10 mm vs. larger) and the number of examined images (transverse vs. both transverse and longitudinal). Then, frequencies of particular sonographic features and EU-TIRADS categories on sUS were compared between the raters. 

Next, the intra-center agreement of nodule classification into particular EU-TIRADS categories and of the evaluation of above-mentioned ultrasound features was assessed. For this purpose, the number of concordant identifications of each category/feature was divided by the number of all diagnoses of that category/feature.

### 2.4. Statistical Evaluation

The statistical analysis was done with the use of Dell Statistica (data analysis software system), version 13, Dell Inc. (2016), Round Rock, TX, USA. The comparison of frequency distributions was performed with the chi-squared test (with modifications appropriate for the number of analyzed cases). The comparison of continuous variables between groups was made with the use of the Kruskal–Wallis test.

The degree of reproducibility in assigning EU-TIRADS categories and identifying sonographic risk features was estimated with Krippendorff’s alpha coefficient (Kα) [20]. Kα ranges from −1 to 1, values close to 1 indicate near perfect agreement, and values above 0.67 are considered an acceptable basis for tentative conclusions. The following interpretation was applied in the study: Kα in the range 0–0.4—poor agreement; 0.41–0.67—fair agreement; 0.68–0.79—substantial agreement; 0.8–1.0—almost perfect agreement. Each EU-TIRADS category and sonographic feature were assessed on a nominal scale. Additionally, an ordinal scale was used for EU-TIRADS categories and some sonographic features where applicable: echogenicity and composition. The value of 0.05 was assumed as the level of significance. 

## 3. Results

### 3.1. Reproducibility of the Assessment on Static US in Relation to Real-Time US

The reproducibility of nodule classification into EU-TIRADS categories on sUS in relation to rtUS, as measured by the percentage of concordant categorizations, was similar for all raters, A: 76.5%, B: 73.8%; C: 70.9%. The reproducibility, as measured with Kα, ranged from 0.60 to 0.68 (Table 2). 

At least 80% of nodules classified into EU-TIRADS category 3 on rtUS were identified as that category on sUS (Table 2). In the case of category 4, the percentage of correct categorizations on sUS was more diverse (from 60.4% to 76.4%). In the case of category 5, a significantly lower percentage of its correct identifications on sUS was noted for rater B than for raters A and C (B: 48.7% vs. A: 73.8% and vs. C: 77.8%, *p* < 0.05 in both cases). On the other hand, category 5 assigned by rater B on sUS was more reliable as it showed the highest percentage of true identifications in relation to all identifications on sUS, B: 60.0% (18 out of 30), A: 51.7% (31 out of 60), C: 40.6% (28 out of 69) (NS) (Table 2 and Table 3). That was a consequence of raters A and C recognizing category 5 on sUS significantly more often than on rtUS (A: 22.4% vs. 15.7%, *p* < 0.05; C: 23.6% vs. 12.3%, *p* < 0.0001). Both those raters more often identified the non-oval shape on sUS (A: 14.2% vs. 8.6%, *p* < 0.05; C: 15.8% vs. 9.3%, *p* < 0.05), and additionally, rater A—marked hypoechogenicity (10.8% vs. 4.9%, *p* < 0.05) (Table 3). 

The analysis of correct identification of high-risk sonographic features on sUS in relation to rtUS showed not a single true recognition of microcalcifications (Table 2). The percentage of correct identification of other high-risk features markedly differed between the raters (especially for marked hypoechogenicity). The reproducibility of high-risk features as measured with Kα was poor or fair for all raters (Table 2). 

We did not find any significant difference in the rate of concordant categorizations in relation to the nodule’s size (≤10 mm vs. larger) nor the number of examined images (single transverse vs. both sections) (Appendix A). 

We did not find any significant difference in the rates of indeterminate or malignant FNA outcomes (categories III–VI) in relation to the method of sonographic image assessment for any EU-TIRADS category. Yet, in the case of EU-TIRADS category 5 established on sUS, the rates of such FNA outcomes were several percentage points lower (from 9.0 to 12.4 points) compared to rtUS, for each rater (Table 4). On the other hand, the rate of benign FNA outcomes was several percentage points higher for two out of three raters A: 40.5% (17 out of 42) vs. 48.3% (29 out of 60); B: 51.4% (19 out of 37) vs. 56.7% (17 out of 3 0); C: 63.9% (23 out of 36) vs. 60.9% (42 out of 69), (NS), sUS vs. rtUS, respectively.

### 3.2. Reproducibility of Nodule Assessment on Static US

Full agreement between three raters in assigning EU-TIRADS category on sUS was observed in 59.3% (499) of nodules (Table 5). The percentage of agreement was similar for each pair of raters, AB: 71.6%, AC and BC: 72.3% in both cases. The reproducibility in assigning EU-TIRADS categories, as measured with Kα, ranged from 0.60 to 0.62; Kα was highest for category 3 and lowest for category 5. The reproducibility of the assessment of sonographic risk features as measured with Kα for all three raters on nominal scale was fair (non-oval shape, hypoechogenicity and macrocalcifications) or poor (other features) (Table 5). Substantial reproducibility was only observed between a pair of raters for macrocalcifications (Kα = 0.69), cystic/spongiform echostructure (Kα = 0.73) and for echogenicity (Kα = 0.70 on ordinal scale). 

In particular, rater A identified marked hypoechogenicity more often than raters B and C (A: 6.8% vs. B: 1.3% and vs. C: 3.3%, *p* < 0.0001 and *p* < 0.005 respectively) and irregular margins more often than rater B (A: 7.6% vs. B: 4.6%, *p* < 0.05) (Appendix A). Rater C was characterized by the highest rates of non-oval shape and microcalcifications and the lowest rate of hypoechogenicity identified in still images. Consequently, rater A classified nodules into category 3 less often than others (A: 34.7% vs. B: 41.9% and vs. C: 41.3%, *p* < 0.005 in both cases), rater B classified nodules into category 5 less often than others (B: 14.3% vs. A: 19.6%, and vs. C: 20.4%, *p* < 0.005 in both cases), while rater C indicated category 4 less often than others (C: 37.3% vs. A: 44.7%, *p* < 0.005 and vs. B: 42.9%, *p* < 0.05) (Appendix A). 

We did not find any significant difference in the rate of categorization agreement in relation to the nodule’s size, nor the number of examined images (Appendix A). Yet, in the case of the most suspected nodules (classified into category 5 on rtUS), we observed that the fully concordant classification by three raters was slightly more often when they analyzed two images than only one image: 50.6% (40 out of 79) vs. 36.1% (13 out of 36), respectively (*p* = 0.1474). In the case of nodules classified into category 3 or 4 on rtUS, the observed differences were smaller and the effect of the number of analyzed static images was reverse. 

Intra-center agreement of nodule classification into particular EU-TIRADS categories on sUS, as measured with the number of fully concordant diagnoses of the category divided by the number of all diagnoses of that category, was significantly lower for category 5 than for categories 3 and 4 (23.2% vs. 50.0% and 43.7%, respectively, both *p* < 0.0001) (Table 6). The agreement of identification of sonographic risk features, determined in the same way, was the highest for solid echostructure (73.7%) and the lowest for microcalcifications (0.0%).

## 4. Discussion

The results obtained in our study showed that the way of sonographic image evaluation influences the nodule’s feature assessment and, eventually, the categorization within EU-TIRADS system. Despite the use of uniform conditions while assessing the nodule’s features—a single ultrasound system, the same probe and preset, dedicated software standardizing the collection of data—we did not obtain a satisfying agreement between the assessment on rtUS and sUS (inter-examination agreement) for any rater. The agreement of EU-TIRADS categorization, as measured with Kα, was close to the level acceptable for tentative conclusion (Kα > 0.67), but it exceeded it only in the case of one rater. Most notable differences between raters in the inter-examination agreement were observed for category 5 (from 48.7% to 77.8% of concordant classifications into that category), less prominent, category 4 (from 60.4% to 76.4%), and the lowest for category 3 (from 80.0% to 86.5%). Remarkably, there was a total disagreement regarding microcalcifications and a strongly variable reproducibility of the identification of marked hypoechogenicity on sUS in relation to rtUS. There was a tendency toward more frequent identification of the non-oval shape and marked hypoechogenicity on sUS. In consequence, identification of the high-risk EU-TIRADS category on sUS seems to be less efficient in the selection of nodules for FNA than its identification on rtUS. 

The inter-rater agreement of the evaluation of images on sUS was also unsatisfactory—Kα for EU-TIRADS categorization was 0.61 for three raters and it was very similar for all possible pairs of raters. The total percentage of concordant categorizations in the EU-TIRADS system was over 70%, but significant differences were noted between particular categories analyzed individually. The lowest agreement (as measured with the ratio between the number of concordant identifications of the category by all raters to the total number of identifications of that category) was observed for category 5 and it was about two times lower than in the case of category 3 or 4. It was a consequence of poor inter-rater reproducibility of three out of four high risk features (i.e., marked hypoechogenicity, microcalcifications and irregular margin, Kα < 0.41 in each case) on sUS. The assessment of nodules on two sections did not improve the inter-exam nor inter-rater reproducibility of nodule categorization in comparison to the assessment on the single section. Yet, in the case of the most suspected nodules (classified into category 5 on rtUS), the rate of fully concordant classification of nodules into category 5 on sUS by three raters was slightly higher when they analyzed two images rather than only one image. 

It is difficult to directly compare our results with data reported by other researchers because of differences in the way of categorization of some sonographic features and various statistical measures of agreement in use [21,22]. The studies concerning the inter-observer agreement of nodule categorization according to EU-TIRADS may be divided into three groups. The first one consists of reports that show values of agreement coefficient similar to our results: Sych et al. [23]—Cohen’s kappa: 0.57, Orhan Soylemez and Gunduz [9]—Cohen’s kappa: 0.65 and Grani et al. [24]—Kα: 0.61. The study by Russ et al. [25] may also be included into that group, although they used an older classification that was a forerunner for EU-TIRADS (Cohen’s kappa: 0.72). The second group of studies—Persichetti et al. [26]; Phuttharak et al. [27]—reports markedly lower values of agreement coefficients (Cohen’s kappa: 0.30 and 0.209, respectively). Persichetti and coworkers admitted that the raters involved in their study had had ‘uneven initial familiarity’ with different classification methodologies [26]. In the latter study, the readers also had variable experience in the examination of thyroid sonograms in the standard manner (2-year or more than 10-year) [27]. The third group of studies—Qi et al. [10]; Hekimosoy et al. [11]—is characterized by almost full agreement in nodule categorization according to EU-TIRADS as found with the intra-class correlation coefficient. However, this coefficient is not quite suited for such analyses. Its applicability is limited to interval or ratio data and it does not account for the agreement by chance. Moreover, it can be used only if there is a normal distribution of examined variable in each measurement or differences between paired measurements are distributed normally [28]—these conditions are not met with regular set of thyroid nodules. In our study, we used the most universal coefficient of agreement—Krippendorff’s alpha. It is useful for nominal, ordinal, or interval data and is suitable for situations with small samples of coded data, multiple raters, or incomplete data [28]. However, it should be kept in mind that its interpretation is much more conservative than in the case of kappa coefficients (Cohen’s kappa or its modification—Fleiss’ kappa). That difference is well illustrated with the exemplary calculation of the Kα and Fleiss’ kappa coefficient in our material for features assessed with a nominal scale for three raters using sUS. Raw values of both coefficients are very similar or—for the non-oval shape, irregular border, solid echostructure and macrocalcifications—even equal (0.58; 0.36; 0.48; 0.65, respectively). Despite the same numeric values, while interpreting Kα, one would find fair or poor agreement in all four cases, but with the use of Fleiss’ kappa, the agreement would be described as moderate, fair, moderate and substantial, respectively. Cohen’s kappa coefficient is more popular, but some researchers indicate its inability to handle situations with largely skewed distributions of codes—e.g., one code appearing much more frequently than another [28]; and such a situation is typical of the assessment of high-risk nodule features that are rarely present, especially in populations with endemic goiter. 

It is important to pay attention to the way in which particular features are categorized when comparing results of agreement studies. The mode of categorization is usually defined by the applied TIRADS. Some features may be assessed on an ordinal scale or a nominal one. The agreement of echogenicity assessment on an ordinal scale was nearly substantial, but when types of echogenicity were dichotomized into two subcategories, then reproducibility of the assessment was variable. Only some researchers considered separately subcategories of echogenicity when analyzing its reproducibility [13,17,29,30]. The same problem refers to the presence of calcifications (the reproducibility for microcalcifications lower than for macrocalcifications) [24,31]. Comparison of the reproducibility of different TIRADS systems is a tricky task due to some specific catches, such as a totally different definition of a feature bearing the same name (feature “shape” according K-TIRADS) or subtle but significant differences in the definition of an apparently similar feature. EU-TIRADS category 5 includes not only taller-than-wide but also round nodules, whereas category 4 encompasses nodules with any hypoechoic part, and not only the dominating one [1]. These subtleties are often missed in studies focused on the comparison of reproducibility of nodule categorization in EU-TIRADS and other systems [9,10]. Despite all these methodological issues, there is a predominant opinion in the majority of reports, concordant with our observations, that the inter-observer agreement is particularly low in the case microcalcifications and irregular borders [12,26,27,29,30,31,32,33,34]. It is not surprising considering the fact that these features intrinsically cannot be reliably assessed on a single image, even optimally selected. Evaluation of entire cine loops should have some advantages over static images in this case [33]. 

The reproducibility of the assessment on sUS in relation to rtUS was evaluated in few studies. This is probably because in the majority of clinics, not all nodule data necessary for such analyses are routinely collected. Bae et al. [17] evaluated nodule classification by K-TIRADS and found a substantial agreement (kappa = 0.75) between rtUS and sUS assessment, with almost perfect agreement for the orientation (a nodule feature defined similarly to the shape in EU-TIRADS) and the lowest agreement for the margin. They generally reported higher kappa values than obtained in our study, which can be explained by two main differences. One is related to the specific attribute of K-TIRADS—it does not distinguish the marked hypoechogenicity as a malignancy feature of high specificity. Another results from different profiles of examined nodules: in our study, nodules with benign cytology predominated what led to relatively low frequency of high risk features, whereas in the study by Bae et al. [17], the majority of nodules were malignant. Yang and Na [35] analyzed the impact of retrospective and prospective US evaluation on the identification of nodule features and the related risk of malignancy, but they did not perform a statistical evaluation of reproducibility of the assessment. They found that the frequencies of microcalcifications, macrocalcifications, irregular margin, comet tail artifact, spongiform appearance were significantly lower and the frequency of solid composition was significantly higher in the retrospective dataset. They also noted that the malignancy risk of solid composition and nonparallel orientation was significantly lower in the retrospective dataset [35]. Comparison between rtUS and sUS was also made by Baek et al. [36] in a study that was actually focused on something else—detection of incidental diffuse thyroid disease. They showed that rtUS was superior to sUS in the diagnostic accuracy; however, there was no significant difference. 

We did not find a decreased reproducibility (either inter-observer or inter-exam) for nodules ≤1 cm. Few researchers studied that problem. Moreover, in some studies, nodules with diameters under 1 cm were excluded [17]. Li et al. [33] in their meta-analysis identified three studies reporting inter-rater agreement for thyroid nodules ≥1 cm and the pooled kappa value was suggested to be higher than the overall kappa value. Park et al. [37] did not find any dependence of inter-observer agreements in the final US assessment on the nodule’s size (ranging from 5 to over 20 mm). 

Our study was intentionally performed in a single center to avoid any additional factors that could potentially affect the obtained results, such as various characteristics of thyroid nodules in examined populations or different profiles of diagnostic centers (endocrine vs. oncological vs. sonographic only). In consequence, our observations need to be confirmed by further studies concerning these potential factors. The study was performed by experienced sonographers and not people trained for the sole purpose of that project, which should be regarded as advantageous. However, this issue is still unsettled as there are contradictory reports on the impact of the sonographer’s experience on the quality of nodule categorization [33,38], with some researchers suggesting that less experienced sonographers rely more on the explicit TIRADS criteria. Our study is also limited by a relatively low frequency of nodules with high-risk features in our material. The majority of nodules were categorized as benign lesions in the FNA outcome. On the other hand, such a distribution of nodules reflects the epidemiologic profile of patients in the clinical practice. Despite the fact that indeterminate or malignant FNA outcomes were relatively rare in our dataset, we noticed that in the case of the EU-TIRADS category 5 established on sUS, the rates of such FNA outcomes were about 10 percentage points lower comparing to rtUS. That observation needs to be confirmed and the actual effect of the method of sonographic feature assessment on the precision of malignancy risk evaluation should be assessed on a set of nodules verified by post-operative histopathological examination. 

## 5. Conclusions

We may conclude that the way of sonographic image evaluation influences the nodule features assessment, the categorization of nodules within EU-TIRADS, and eventually, the selection of nodules for FNA. Further studies are necessary to find out how that translates into the effectiveness of identifying cancers as verified by post-operative examination. The assessment of nodule sonographic features on sUS with the use of stored images has an unsatisfactory inter-rater reproducibility even when the method of assessment is standardized and it is performed by sonographers well experienced in thyroid US imaging, working at the same diagnostic center. Better inter-observer and inter-exam agreement may be obtained with the use of stored video clips that show the whole volume of the nodule, but this should be confirmed in future studies. Subjectivity of sonographic evaluation of thyroid nodules may be reduced with the introduction of computer-aided diagnostic systems, and particularly artificial intelligence-assisted ultrasonography [39]; however, further research is needed to better explore these diagnostic techniques. Undoubtedly, studies focused on the comparison of effectiveness of various TIRADS should clearly describe the method of sonographic image evaluation used.

## Figures and Tables

**Table 1 jcm-12-05809-t001:** Characteristics of the examined set of thyroid nodules.

	EU-TIRADS Category Assigned on Real-Time Sonography				
	EU-TIRADS 2	EU-TIRADS 3	EU-TIRADS 4	EU-TIRADS 5	All Nodules
**No./% of nodules**	8/9.5%	299/35.5%	420/49.9%	115/13.7%	842
**No./% of patients**	4/0.6%	186/29.0%	343/53.5%	108/16.8%	641
**No./% of males**	1/25.0%	29/15.6%	41/12.0%	15/13.9%	86
**Age of patients, mean ± SD [year]**	58.0 ± 18.7 ^a^	68.4 ± 14.2	62.6 ± 12.4	59.9 ± 14.8 ^a^	64.2 ± 13.8
**Volume of nodules, mean ± SD [cm^3^]**	8.4 ± 9.5	4.7 ± 10.4	2.9 ± 11.4	2.9 ± 13.6	3.6 ± 11.1
**No./% of nodules < 1 cm**	0	18/6.0% ^b^	97/23.1%	36/31.3%	151
**No./% of single image (transverse section)**	3/1.0%	110/37.0%	148/49.8%	36/12.1%	297
**No./% of two images (both sections)**	5/0.9%	189/34.7%	272/49.9%	79/14.5%	545
**Outcomes of FNAs [No./%]**					
**Bethesda category I**	7/87.5% ^c^	45/15.1%	73/17.4%	18/15.7%	143
**Bethesda category II**	1/12.5% ^de^	224/74.9%	255/60.7%	59/51.3%	539
**Bethesda category III**	0	30/10.0%	81/19.3%	25/21.7%	136
**Bethesda category IV**	0	0	2/0.5%	0	2
**Bethesda category V**	0	0	5/1.2%	8/7.0%	13
**Bethesda category VI**	0	0	4/1.0%	5/4.4%	9

^a^—*p* < 0.05 vs. EU-TIRADS 3; ^b^—*p* < 0.0001 vs. EU-TIRADS 4 and EU-TIRADS 5; ^c^—*p* < 0.0001 vs. EU-TIRADS 3, EU-TIRADS 4 and EU-TIRADS 5; ^d^—*p* < 0.0001 EU-TIRADS 3; ^e^—*p* < 0.05 vs. EU-TIRADS 4.

**Table 2 jcm-12-05809-t002:** Reproducibility of EU-TIRADS categorization and the assessment of sonographic features on static US in relation to real-time US as measured by the percentage of true positive assignments of each category/feature on static US and Krippendorff’s alpha coefficient.

Category/Feature	% (No.) of True Positive Cases on Static US in Relation to Real-Time US				Krippendorff’s Alpha Coefficient		
	Rater A	Rater B	Rater C	*p*	Rater A	Rater B	Rater C
**All EU-TIRADS** **categories**	76.5 (205)	73.8 (208)	70.9 (207)	NS	0.68	0.60	0.64
**EU-TIRADS 2**	50.0 (1)	33.3 (1)	100.0 (3)	NS	0.33	0.40	1.0
**EU-TIRADS 3**	80.0 (40)	83.1 (133)	86.5 (77)	NS	0.70	0.60	0.66
**EU-TIRADS 4**	76.4 (133)	68.3 (56)	60.4 (99)	<0.005 A vs. C	0.56	0.51	0.48
**EU-TIRADS 5**	73.8 (31)	48.7 (18)	77.8 (28)	<0.05 B vs. A, C	0.52	0.48	0.43
**marked hypoechogenicity**	84.6 (11)	12.5 (1)	25.0 (1)	<0.01 A vs. B	0.48	0.17	0.14
**non-oval shape**	65.2 (15)	47.1 (8)	77.8 (21)	<0.05 B vs. C	0.43	0.47	0.52
**irregular margins**	56.3 (9)	46.2 (6)	30.0 (3)	NS	0.41	0.46	0.16
**microcalcifications**	0.0 (0)	0.0 (0)	0.0 (0)	NS	−0.01	−0.01	−0.01
**hypoechogenicity**	82.7 (163)	79.8 (79)	75.6 (136)	NS	0.63	0.61	0.60
**solid**	87.8 (180)	84.6 (214)	98.7 (226)	<0.0001 C vs. A, B	0.63	0.50	0.39
**cystic/spongiform**	50.0 (1)	66.7 (2)	100.0 (3)	NS	0.33	0.57	1.0
**macrocalcifications**	66.7 (16)	47.4 (9)	58.1 (18)	NS	0.69	0.38	0.66
**echogenicity** **(ordinal scale)**	83.6 (224)	81.2 (229)	79.5 (232)	NS	0.71	0.66	0.62
**composition** **(ordinal scale)**	83.2 (223)	83.3 (235)	83.9 (245)	NS	0.65	0.53	0.41

**Table 3 jcm-12-05809-t003:** Frequencies of particular categories of EU-TIRADS and sonographic features on real-time US and static US as assigned by raters A, B, and C (rater A—268 nodules, rater B—282 nodules, rater C—292 nodules).

Category/ Feature	Rater A			Rater B			Rater C		
	Real-Time % (No.)	Static % (No.)	*p*	Real-Time % (No.)	Static % (No.)	*p*	Real-Time % (No.)	Static % (No.)	*p*
**EU-TIRADS 2**	0.7 (2)	1.5 (4)	NS	1.1 (3)	0.7 (2)	NS	1.0 (3)	1.0 (3)	NS
**EU-TIRADS 3**	18.7 (50)	20.5 (55)	NS	56.7 (160)	57.1 (161)	NS	30.5 (89)	37.7 (110)	NS
**EU-TIRADS 4**	64.9 (174)	55.6 (149)	<0.05	29.1 (82)	31.6 (89)	NS	56.2 (164)	37.7 (110)	<0.0001
**EU-TIRADS 5**	15.7 (42)	22.4 (60)	<0.05	13.1 (37)	10.6 (30)	NS	12.3 (36)	23.6 (69)	<0.0001
**marked hypoechogenicity**	4.9 (13)	10.8 (29)	<0.05	2.8 (8)	0.7 (2)	NS	1.4 (4)	3.1 (9)	NS
**non-oval shape**	8.6 (23)	14.2 (38)	<0.05	6.0 (17)	6.7 (19)	NS	9.3 (27)	15.8 (46)	<0.05
**irregular margins**	6.0 (16)	9.0 (24)	NS	4.6 (13)	4.3 (12)	NS	3.4 (10)	6.9 (20)	NS
**microcalcifications**	1.1 (3)	1.9 (5)	NS	1.8 (5)	0.7 (2)	NS	0.7 (2)	2.4 (7)	NS
**hypoechogenicity**	73.5 (197)	64.2 (172)	<0.05	35.1 (99)	39.4 (111)	NS	61.6 (180)	51.4 (150)	NS
**solid composition**	76.5 (205)	72.0 (193)	NS	89.7 (253)	76.2 (215)	<0.0001	78.4 (229)	91.8 (268)	<0.0001
**cystic/spongiform**	0.8 (2)	1.5 (4)	NS	1.0 (3)	1.4 (4)	NS	1.0 (3)	1.0 (3)	NS
**macrocalcifications**	9.0 (24)	7.8 (21)	NS	6.7 (19)	8.2 (23)	NS	10.6 (31)	7.2 (21)	NS

**Table 4 jcm-12-05809-t004:** Comparison of the frequency of indeterminate or malignant FNA outcome in relation to EU-TIRADS category as assigned with both methods of US image evaluation (real-time vs. static).

Category	Rater A		Rater B		Rater C	
	Real-Time [%]	Static [%]	Real-Time [%]	Static [%]	Real-Time [%]	Static [%]
**EU-TIRADS 2**	0.0 (0 out of 2)	0.0 (0 out of 4)	0.0 (0 out of 3)	0.0 (0 out of 2)	0.0 (0 out of 3)	0.0 (0 out of 3)
**EU-TIRADS 3**	14.0 (7 out of 50)	12.7 (7 out of 55)	11.9 (19 out of 160)	11.2 (18 out of 161)	4.5 (4 out of 89)	10.9 (12 out of 110)
**EU-TIRADS 4**	24.1 (42 out of 174)	28.2 (42 out of 149)	20.7 (17 out of 82)	27.0 (24 out of 89)	20.1 (33 out of 164)	20.0 (22 out of 110)
**EU-TIRADS 5**	38.1 (16 out of 42)	26.7 (16 out of 60)	32.4 (12 out of 37)	20.0 (6 out of 30)	27.8 (10 out of 36)	18.8 (13 out of 69)

No statistically significant differences were observed.

**Table 5 jcm-12-05809-t005:** Agreement in assigning EU-TIRADS category and nodule’s sonographic features between particular raters evaluating static US as measured by percentage of concordant assignments, including positive and negative cases (p/n), and Krippendorff’s alpha coefficient—analysis of 842 nodules.

Category/ Feature	Rater	% (p/n) of Concordant Assignments			Krippendorff’s Alpha Coefficient		
		A	B	All	A	B	All
**All EU-TIRADS categories ***	**B**	71.6 (603)	-	59.3 (499)	0.61	-	0.61
	**C**	72.3 (609)	72.3 (609)		0.62	0.60	
**EU-TIRADS 2**	**B**	98.7 (3/828)	-	98.6 (3/827)	0.35	-	0.52
	**C**	98.9 (4/829)	99.5 (6/832)		0.47	0.75	
**EU-TIRADS 3**	**B**	83.5 (253/450)	-	73.4 (224/394)	0.65	-	0.63
	**C**	81.2 (241/443)	82.1(275/416)		0.60	0.63	
**EU-TIRADS 4**	**B**	77.3 (273/378)	-	67.2 (214/352)	0.54	-	0.55
	**C**	78.4 (254/406)	78.7 (248/415)		0.55	0.56	
**EU-TIRADS 5**	**B**	83.7 (74/631)	-	77.1 (58/591)	0.42	-	0.48
	**C**	86.1 (110/615)	84.3 (80/630)		0.57	0.45	
**marked hypoechogenicity**	**B**	93.5 (7/787)	-	91.6 (3/768)	0.17	-	0.24
	**C**	93.5 (15/772)	96.2 (4/806)		0.32	0.19	
**non-oval shape**	**B**	90.7 (59/705)	-	86.7 (47/683)	0.55	-	0.58
	**C**	91.7 (74/698)	91.0 (62/704)		0.63	0.57	
**irregular border**	**B**	92.0 (18/757)	-	89.2 (12/739)	0.31	-	0.36
	**C**	92.9 (26/756)	93.5 (16/771)		0.43	0.33	
**microcalcifications**	**B**	98.2 (0/827)	-	97.3 (0/819)	−0.01	-	0.27
	**C**	98.1 (6/820)	98.2 (3/824)		0.42	0.28	
**hypoechogenicity**	**B**	81.1 (365/318)	-	70.4 (297/296)	0.62	-	0.61
	**C**	77.2 (334/316)	82.5 (4//347)		0.54	0.65	
**solid composition**	**B**	85.2 (570/147)	-	75.8 (569/69)	0.60	-	0.48
	**C**	77.6 (582/71)	88.8 (679/69)		0.29	0.53	
**cystic/spongiform**	**B**	98.7 (5/826)	-	98.6 (5/825)	0.47	-	0.60
	**C**	99.0 (6/828)	99.4 (7/830)		0.60	0.73	
**macrocalcifications**	**B**	95.1 (53/748)	-	92.4 (34/744)	0.69	-	0.65
	**C**	95.5 (37/767)	94.2 (47/746)		0.64	0.63	
**echogenicity *** **(ordinal scale)**	**B**	80.4 (677)	-	69.5 (585)	0.70	-	0.67
	**C**	76.6 (645)	81.6 (687)		0.65	0.67	
**composition *** **(ordinal scale)**	**B**	82.7 (696)	-	73.6 (620)	0.62	-	0.51
	**C**	75.9 (639)	88.0 (741)		0.32	0.55	

*—positive cases only.

**Table 6 jcm-12-05809-t006:** Intra-center agreement in the classification of nodules into particular EU-TIRADS categories and in assigning their sonographic features on static US as defined by the ratio of the number of concordant identification of each category/feature to the number of all assignments of that category/feature and expressed as a percentage.

Category/Feature	Intra-Center Agreement % (No.)	*p*
**EU-TIRADS 2**	20.0 (3 out of 15)	
**EU-TIRADS 3**	50.0 (224 out of 448)	<0.0001 vs. EU-TIRADS 5
**EU-TIRADS 4**	43.7 (214 out of 490)	<0.0001 vs. EU-TIRADS 5
**EU-TIRADS 5**	23.2 (58 out of 250)	
**marked hypoechogenicity**	4.1 (3 out of 73)	<0.0001 vs. non-oval shape and macrocalcifications
**non-oval shape**	29.6 (47 out of 159)	<0.001 vs. irregular margins <0.05 vs. microcalcifications
**irregular border**	11.6 (12 out of 103)	<0.005 vs. macrocalcifications
**microcalcifications**	0.0 (0 out of 23)	<0.005 vs. macrocalcifications
**hypoechogenicity**	54.4 (297 out of 546)	<0.005 vs. macrocalcifications <0.05 vs. cystic/spongiform <0.0001 vs. all other features
**solid composition**	73.6 (569 out of 773)	<0.05 vs. cystic/spongiform <0.0001—vs. all other features
**cystic/spongiform**	62.5 (5 out of 8)	
**macrocalcifications**	34.7 (34 out of 98)	

## Data Availability

The data presented in this study are available on request from the corresponding author. The data are not publicly available due to patient privacy restrictions.

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
