# Peer review of "Real-Time Ultrasonography and the Evaluation of Static Images Yield Different Results in the Assessment of EU-TIRADS Categories"

_jcm, 2023, doi:10.3390/jcm12185809_

Round 1

Reviewer 1 Report

The study you provided appears to have a well-structured design that aims to investigate the impact of different methods of sonographic image evaluation on the categorization of thyroid nodules within the EU-TIRADS system. Here are some aspects of the study design that contribute to its strengths and considerations:

Strengths:

- Clear Objectives: The study's objectives are well-defined, focusing on comparing the results of nodule categorization within the EU-TIRADS system based on two different methods of sonographic image evaluation: real-time ultrasonography (rtUS) and static ultrasonography (sUS).

- Controlled Setting: The study was conducted in a single center with over 20 years of experience in thyroid ultrasound and fine-needle aspiration (FNA), contributing to consistency in imaging and assessment practices.

- Experienced Raters: The study was conducted by three experienced doctors with over 15 years of experience in performing ultrasound and FNA of the thyroid gland. This expertise adds credibility to the evaluation process.

- Large Sample Size: The study included 842 nodules in 641 patients, providing a substantial dataset for analysis.

- Detailed Data Collection: The study employed a custom computer program to collect detailed information on examined nodules, including their location, echogenicity, dimensions, and relevant sonographic features. This standardized approach helps ensure data accuracy and consistency.

- Comparison of Multiple Features: The study assessed various sonographic features, such as echogenicity, composition, shape, margins, and calcifications, allowing for a comprehensive evaluation of their impact on nodule categorization.

- Statistical Analysis: The study used appropriate statistical methods, including Krippendorff's alpha coefficient, to measure inter-rater agreement and reproducibility of nodule categorization.

I have some considerations to highlight:

The study included data collected by experienced doctors, which could introduce bias if these doctors are exceptionally skilled or have specific preferences for certain classifications. 

The study was conducted in a single center, which might limit the generalizability of the findings to other healthcare settings with different patient populations and levels of expertise.

The study mentions a relatively low frequency of nodules with high-risk features, which could affect the precision of evaluating the impact of those features on categorization, and did not find a significant impact of nodule size on inter-rater agreement. However, nodule size might affect the assessment of certain features (e.g., microcalcifications), and this aspect could be explored further.

While the study investigates the impact of different assessment methods on nodule categorization, it doesn't directly address how these differences might impact clinical decision-making or patient outcomes.

Overall, the study's design appears to be comprehensive and well-suited to address the research question. 

Congratulations to the authors.

Author Response

The study you provided appears to have a well-structured design that aims to investigate the impact of different methods of sonographic image evaluation on the categorization of thyroid nodules within the EU-TIRADS system. Here are some aspects of the study design that contribute to its strengths and considerations:

 Strengths:

- Clear Objectives: The study's objectives are well-defined, focusing on comparing the results of nodule categorization within the EU-TIRADS system based on two different methods of sonographic image evaluation: real-time ultrasonography (rtUS) and static ultrasonography (sUS).

- Controlled Setting: The study was conducted in a single center with over 20 years of experience in thyroid ultrasound and fine-needle aspiration (FNA), contributing to consistency in imaging and assessment practices.

- Experienced Raters: The study was conducted by three experienced doctors with over 15 years of experience in performing ultrasound and FNA of the thyroid gland. This expertise adds credibility to the evaluation process.

- Large Sample Size: The study included 842 nodules in 641 patients, providing a substantial dataset for analysis.

- Detailed Data Collection: The study employed a custom computer program to collect detailed information on examined nodules, including their location, echogenicity, dimensions, and relevant sonographic features. This standardized approach helps ensure data accuracy and consistency.

- Comparison of Multiple Features: The study assessed various sonographic features, such as echogenicity, composition, shape, margins, and calcifications, allowing for a comprehensive evaluation of their impact on nodule categorization.

- Statistical Analysis: The study used appropriate statistical methods, including Krippendorff's alpha coefficient, to measure inter-rater agreement and reproducibility of nodule categorization.

We are thankful for kind words about our manuscript and the enumeration of its strengths.

I have some considerations to highlight:

The study included data collected by experienced doctors, which could introduce bias if these doctors are exceptionally skilled or have specific preferences for certain classifications. 

Indeed, the study was carried out by three doctors with at least 15 years of practice in the same diagnostic center, very experienced in performing ultrasound imaging and FNA of the thyroid. Of course, we realize that this is not a typical composition of the diagnostic team, which usually varies in experience. However, since the impact of the experience on the concordance of nodule classification into TIRADS categories is not clear, we decided that selecting a team with similar experience gained in a single center would allow to avoid an additional factor that could affect the reproducibility of the results. There are reports whose authors claim that experience is a necessary factor for the TIRADS assessment to be credible. But some other researchers suggested that less experienced sonographers follow the recommendations/recommendations more rigorously, while those with extensive experience have their own habits and are – in consequence - less flexible in adapting to the recommendations. However, this issue was not the subject of our analyses and we indicated that accordingly as a limitation to our study at lines 383-386 of the original manuscript.

The study was conducted in a single center, which might limit the generalizability of the findings to other healthcare settings with different patient populations and levels of expertise.

We do agree with the view that the characteristics of examined population of patients affects the results of classification of nodules into TIRADS categories and that effect should not be disregarded. But it is safe to assume that the regularities observed by us might be even more distinct in some other populations. Our patients came from population that had been exposed to iodine deficiency for many years and thus their thyroid nodules relatively rarely showed high risk features. We indicated that issue at lines 373-375 of the paragraph concerning limitations to our study. Despite that limitation we managed to observe that in the case of EU-TIRADS category 5 established on sUS the rates of indeterminate or malignant FNA outcomes were  lower comparing to rtUS. But due to limited number of category 5 nodules we stipulated that further research is needed to confirm this observation – line 379. We also agree that the reference level of the center and its profile, e.g. endocrine vs. oncological, is important. We investigated this issue in relation to the reproducibility of Bethesda Category III formulation (SÅ‚owiÅ„ska-Klencka et al. Low reproducibility of equivocal categories of the Bethesda System for Reporting Thyroid Cytology makes the associated risk of malignancy specific to the diagnostic center. Endocrine. 2021, 74, 355-364. doi: 10.1007/s12020-021-02781-3.). In the paragraph concerning limitations of our study, we added a sentence that summarized these two issues:

”Our study was intentionally performed in a single center to avoid any additional factors that could potentially affect obtained results, such as various characteristics of thyroid nodules in examined populations or different profiles of diagnostic centers (endocrine vs oncological vs sonographic only). In consequence, our observations need to be confirmed by further studies concerning these potential factors”

The study mentions a relatively low frequency of nodules with high-risk features, which could affect the precision of evaluating the impact of those features on categorization, and did not find a significant impact of nodule size on inter-rater agreement. However, nodule size might affect the assessment of certain features (e.g., microcalcifications), and this aspect could be explored further.

As we have already mentioned above, we raised the issue of low frequency of nodules with high-risk features in our material while enumerating limitations of the study. Therefore, we did not separately assess the effect of nodule size on the reproducibility of individual high-risk features, but only on the classification of nodules into particular EU-TIRADS categories. We agree that such an impact may vary for particular high-risk features and requires further research.

While the study investigates the impact of different assessment methods on nodule categorization, it doesn't directly address how these differences might impact clinical decision-making or patient outcomes.

Our study was focused on the problem of the possible influence of the ultrasound image assessment method on the categorization of thyroid nodules. That was a necessary first step confirming the existence of such an effect. The conclusions of our study are important because they draw attention to a problem that is often overlooked by the researchers – that is, the issue of an accurate description of the ultrasound image assessment method, on the basis of which a study dedicated to some clinically important aspects of the diagnosis of thyroid nodules is constructed. The study concentrated on the impact of the ultrasound image assessment method on clinical decision-making would be a natural second stage of research, but it requires a different methodological structure and should be based on the set of nodules verified with the results of postoperative histopathological examinations or long-term clinical assessment. We mentioned the necessity of such studies at lines 379-381 of the original submission.

But inspired by this Reviewer’s comment we emphasized in the revision the preliminary data on the lower efficiency of sUS than rtUS in the selection of nodules for FNA. To this end we moved the table showing the related data from supplementary material s to the main manuscript and we added a sentence describing more thoroughly that issue after line 216 of the original and another related sentence to the discussion at line 271 of the original.

Reviewer 2 Report

General comment:

This a very interesting study to look at the performance differences between real-time reading of thyroid ultrasound scan and reading of saved ultrasound images. The study findings would have important clinical implications in second-opinion review of the ultrasound images and in the performance of computer-aided diagnosis (CAD) software, which is mostly provided to read static images. The collected database of the study is also sufficient with 3 experienced raters and 842 nodules. However, the authors have failed to answer the critical research questions and to discuss the critical clinical implication of the study findings. Major revisions are required to make clear the contribution of the study.

Major comments:

1.      The most important research question should be how the static image review differs from the real-time scan reading in recommending FNA since this is sole purpose of EU-TIRADS and the most critical in clinical practice of managing thyroid nodules. The authors have failed to answer the question throughout the manuscript though all the nodules have FNA results. Instead, the FNA results were only shown in the supplement and roughly classified into indeterminate/malignant and benign groups without further breaking down into the details. The risk of being FNA indeterminate/malignant by the real-time readings and static-image review should be further analyzed and discussed.

2.      The analysis of the study results should focus on how the static image reading affects the second-opinion review, by a CAD software device or by another physician, and/or future follow-up by the same physician. For example, are more potential thyroid cancers (indeterminate/malignant FNA results) missed by the static image reviews or are more benign nodules over-diagnosed by the static image reviews?

3.      Since the FNA recommendation is also based on the nodule size, the size information, i.e., <1cm, [1cm, 1.5cm), [1.5cm, 2cm), >=2cm, should be provided together with their EU-TIRADS and FNA results. Only with the size information, the analysis of how the static image review differs on the FNA recommendation is possible.

Detailed comments:

1.      The FNA results should be included in the demographic data (Table 1).

2.      The reproducibility of nodule assessment should be characterized as follow-up static image review by the same physician and analyzed under different FNA outcomes (Table 2 and Table S1). Also, Table S1 appears to me more comprehensible than Table 2. Perhaps, Table 2 should be placed in the supplement instead.

3.      Table 3 is far more clinically insignificant then Table S2 and is suggested to be replaced by Table S2.

4.      Table 4 should be re-arranged to characterize the scenario of second-opinion image review by different physicians and should also be analyzed under different FNA outcomes.

5.      It is unclear how the positive and negative cases are defined and what the % (p/n) means in Table 4. It is difficult to comprehend why the marked hypo-echogenicity has concordant assignment of above 90% and its Ka is only 0.17-0.32. The same can be observed for the irregular border and microcalcifications.

6.      References of second-opinion static image reviews, e.g. for breast cancers, should be added.

no comments

Author Response

General comment:

This a very interesting study to look at the performance differences between real-time reading of thyroid ultrasound scan and reading of saved ultrasound images. The study findings would have important clinical implications in second-opinion review of the ultrasound images and in the performance of computer-aided diagnosis (CAD) software, which is mostly provided to read static images. The collected database of the study is also sufficient with 3 experienced raters and 842 nodules. However, the authors have failed to answer the critical research questions and to discuss the critical clinical implication of the study findings. Major revisions are required to make clear the contribution of the study.

We understand intentions of the Reviewer, who proposes to pose further research questions. However, our study in its current form is extensive, contains many results and important observations. We are aware that there are very interesting publications confirming the high clinical usefulness of second-opinion review, when it is performed by an experienced doctor both on the basis of real-time assessment and on the basis of recorded photos and / or videos from ultrasound examination. However, our current work focuses on a different problem. It is devoted to the assessment of inter-exam and inter-observer reproducibility of the assessment of ultrasound features of thyroid nodules and their categorization by raters with similar, long-time experience, working at a single center. The conclusions of our work clearly indicate that this reproducibility is not satisfactory, and the categorization of nodules based on one or two recorded images does not reflect the categorization obtained during real-time. This observation indicates the need for a thorough description of the method of assessing the ultrasound image of the nodule in studies devoted to the assessment of the clinical usefulness of TIRADS systems, which, as we have shown in the discussion, is rare and hinders reliable conclusions. In our opinion, the results obtained by us show an important but underestimated methodological problem which affects clinical practice. They suggest that the identification of the high risk EU-TIRADS category on sUS is less efficient in selection of nodules for FNA than on rtUS. We emphasized that observation in the revision. This issue was elaborated in more detail in the  answers to major comments.

Major comments:

  1. The most important research question should be how the static image review differs from the real-time scan reading in recommending FNA since this is sole purpose of EU-TIRADS and the most critical in clinical practice of managing thyroid nodules. The authors have failed to answer the question throughout the manuscript though all the nodules have FNA results. Instead, the FNA results were only shown in the supplement and roughly classified into indeterminate/malignant and benign groups without further breaking down into the details. The risk of being FNA indeterminate/malignant by the real-time readings and static-image review should be further analyzed and discussed.

We absolutely agree with the Reviewer that the question how the static image review differs from the real-time scan reading in regard to the selection of nodules for FNA is extremely important and that is primary question which arises from the results of our study. Notably, most of our recent publications are devoted to the assessment of the clinical effectiveness of various systems for stratification of ultrasound risk of thyroid nodules in selecting lesions for FNA and revealing cancers, also taking into account additional clinical aspects, such as the coexistence of Hashimoto's disease. However, solid conclusions in this area always require not only analysis of FNA results, but, above all, their verification against final postoperative histopathological diagnoses. Thyroid FNA outcomes are often indeterminate (categories III-V Bethesda classification), and the risk of malignancy for different categories of inconclusive outcomes varies greatly between centers. Only on the basis of final histopathological diagnoses or on the basis of long-term clinical assessment will it be possible to find out whether there are differences in the clinical effectiveness of both methods of ultrasound image assessment. We are currently collecting data for such analysis. For the above reasons, in the present study we considered the results of microscopic evaluation based only on cytological material with caution and therefore we included them in supplementary materials and did not formulate any strong conclusion based on these preliminary observations. We are aware that these results are interesting because they suggest that static US evaluation has less clinical value than real-time US and is less likely to lead to indeterminate or malignant cytology when EU-TIRADS category 5 is assigned. However, these observations must be confirmed on a larger group of nodules with a known final histopathological diagnosis. Especially as in many centers, unlike in ours, it is static US that is the basis for research on the categorization of nodules into various TIRADS systems. However, inspired by the Reviewer's remark, we decided to refer to this issue a little more in our paper and to move Table S2 showing the comparison of the frequency of indeterminate or malignant FNA outcome in relation to EU-TIRADS category as assigned with both methods of US image evaluation to the main part of the paper. In the Results (after line 216 of the original submission) we added a sentence describing that problem in more detail not only in relation to indeterminate/malignant cytology, but also to benign cytology. Similarly, the issue was also addressed while summarizing obtained results in the Discussion (after line 271 of the original).

  1. The analysis of the study results should focus on how the static image reading affects the second-opinion review, by a CAD software device or by another physician, and/or future follow-up by the same physician. For example, are more potential thyroid cancers (indeterminate/malignant FNA results) missed by the static image reviews or are more benign nodules over-diagnosed by the static image reviews?

Again, we want to emphasize that the purpose of our study was not to check whether a desirable change in the categorization of the nodule assigned on real-time US can be made based on static US. It is quite reasonable to assume that it can, especially when real-time US performed by inexperienced sonographer is verified by an expert using static images. We investigated the inverse relationship – that is, whether the assessment of the image on static US gives a chance to reconstruct the assessment of the nodule’s appearance formulated during real-time US. We assumed that with similar, high experience of raters, real-time US carries more information than one or two recorded static pictures. This assumption seems to be supported by our observation that more frequent recognition of category 5 of EU-TIRADS on static US than real-time US is not associated with more frequent alarming cytology from such nodules.

As we have already explained, it is difficult to reliably determine the number of potential missed thyroid cancers based on the FNA outcome alone, without knowing the final histopathological diagnosis. This would be burdened with a large error, especially in the case of category III nodules, due to the very different risk of its malignancy depending on the type of atypia (nuclear vs. other). Nevertheless, in the original version of the results in lines 212-216 we gave the most important preliminary information:

„We didn’t find any significant difference in the rates of indeterminate or malignant FNA outcomes (categories III-VI) in relation to the method of sonographic image assessment for any EU-TIRADS category. But in the case of EU-TIRADS category 5 established on sUS the rates of such FNA outcomes were several percentage points lower (from 9.0 to 12.4 points) comparing to rtUS, for each rater.”

As it has been mentioned, the information on differences in rates of benign FNA outcomes in nodules of EU-TIRADS category 5 as identified on sUS or rtUS was added to the revision. We also extended Table 1 to show FNA outcomes of all examined nodules.

  1. Since the FNA recommendation is also based on the nodule size, the size information, i.e., <1cm, [1cm, 1.5cm), [1.5cm, 2cm), >=2cm, should be provided together with their EU-TIRADS and FNA results. Only with the size information, the analysis of how the static image review differs on the FNA recommendation is possible.

We agree with the Reviewer's opinion that when examining the impact of various methods of ultrasound image assessment on the effectiveness of the selection of thyroid nodules to FNA, it is also necessary to take into account the size of the nodule, which is an important parameter defined by EU-TIRADS for each category. However, as we have already mentioned, this issue will be the subject of a separate study, which requires knowledge of the results of postoperative histopathological examinations of nodules. 

Detailed comments:

  1. The FNA results should be included in the demographic data (Table 1).

Table has been extended accordingly.

  1. The reproducibility of nodule assessment should be characterized as follow-up static image review by the same physician and analyzed under different FNA outcomes (Table 2 and Table S1). Also, Table S1 appears to me more comprehensible than Table 2. Perhaps, Table 2 should be placed in the supplement instead.

Table 2 directly answers the question whether static US allows to reconstruct the conclusions that were formulated during the real-time US. Table S1 shows data on the frequency of identification of particular features by each rater using both methods, and although it is easier to comprehend, it does not contain key variables describing the reproducibility of the assessment, so we would prefer to leave Table 2 in the main part of the paper and just add the original Table S1 to it.

In the current study, we do not intend to relate the reproducibility TIRADS categorization to FNA results, as we believe that it will be more reliable to relate that reproducibility to the results of postoperative histopathological examination, which, as already mentioned, we are currently working on.

  1. Table 3 is far more clinically insignificant then Table S2 and is suggested to be replaced by Table S2.

As suggested by the Reviewer, Table S2 was moved to the main part of the paper and Table 3 was placed in supplementary materials.

  1. Table 4 should be re-arranged to characterize the scenario of second-opinion image review by different physicians and should also be analyzed under different FNA outcomes.

As we wrote above, we do not plan to relate the reproducibility of TIRADS categorization to FNA results in this study for two reasons – first, it would greatly complicate the structure of Table 4 (5 in the revised version), which in the current version already occupies almost the entire page. Second, we believe that such an analysis should also refer to the results of postoperative histopathological examination and thus it requires a separate study.

  1. It is unclear how the positive and negative cases are defined and what the % (p/n) means in Table 4. It is difficult to comprehend why the marked hypo-echogenicity has concordant assignment of above 90% and its Ka is only 0.17-0.32. The same can be observed for the irregular border and microcalcifications.

The percentage agreement of the determination of any ultrasound feature includes both positive cases and negative cases. Both of these values are given in this table precisely to make it easier to understand why in some cases, with almost the same percentage agreement in determining a given characteristic, there are significant differences in the values of Krippendorff's alpha coefficient. This can be clearly seen on the example of microcalcification assessment. Below there is an excerpt from Table 4 relating to this feature. Microcalcifications were rarely found in our material. Raters often agreed that there were no calcifications in the nodule, e.g. rater A and rater B in 827 cases agreed in the opinion that there were no microcalcifications (827 negative cases), but in no case indicated the presence of microcalcifications in the same nodule (0 positive cases). In turn, raters A and C in 820 cases agreed that there were no microcalcifications, but in 6 cases they concordantly indicated their presence in the same nodule.  As a result, Krippendorff's alpha coefficient was significantly higher for AC than AB. Both pairs had a similar number of consistent opinions (827 and 826, respectively), but only the second pair unanimously indicated the presence of microcalcifications (positive cases). Percent agreement does not take into account dataset imbalance in contrast to Krippendorff's alpha coefficient that calculates the ratio between the expected and observed percent agreement.

In general, lower than intuitively expected values of Krippendorff’s alpha coefficient are observed when there are largely skewed distributions of codes —e.g., one code appearing much more frequently than another.

Table 4. Agreement in assigning EU-TIRADS category and nodule’s sonographic features between particular raters evaluating static US as measured by percentage of concordant assignments, including positive and negative cases (p/n), and Krippendorff’s alpha coefficient – analysis of 842 nodules.

category/

feature

rater

% (p/n) of concordant assignments

Krippendorff’s alpha coefficient

A

B

All

A

B

all

Microcalcifications

B

98.2 (0/827)

-

97.3 (0/819)

-0.01

-

0.27

C

98.1 (6/820)

98.2 (3/824)

0.42

0.28

  1. References of second-opinion static image reviews, e.g. for breast cancers, should be added.

We are familiar with these interesting reviews but we find it difficult to relate them to our study, which aim wasn’t to check whether a desirable change in the categorization of the nodule assigned on real-time US can be made based on static US. Especially as we believe that a great part of the clinical value of second-opinion static image review is connected with the second opinion itself. The question whether the second-opinion would benefit from being based on real-time reassessment rather than a review of static images seems very interesting in this respect but it definitely is out of scope of our study.

Reviewer 3 Report

Major Concerns:

  1. Please provide the general characteristics of the study population, including sex, age, mean thyroid nodule size, the exact number of patients in each TIRADS group, and the specific Bethesda classification of the nodules, not just the malignant FNA outcome (III-IV). This will allow the reader to have an overview of the enrolled patients.
  2. In Table 3, the number of examined images (single transverse vs. both sections) did not affect the concordant rate of TIRADS categorization. Moreover, in rater A and C, the concordant rate is even higher in the single image group than the two-image group. Is there any possible explanation for this result? Theoretically, two images should contain more information and have a higher concordant rate compared to a single image.
  3. Following the above question, is it possible that the study only enrolled patients who have two static images (n=545)? Simplifying the study by focusing on patients with two static images might make it easier to understand. Additionally, the concordant rate of assessing TIRADS in patients with different numbers of static images (1 vs. 2) could still be evaluated by giving the raters only one static image (transverse section) first, obtaining the first TIRADS classification, and then giving them the second still image (longitudinal section) for re-evaluation. If this cannot be done, please address this point in the limitations section.
  4. Please specify the three raters' experience in performing ultrasound and fine needle aspiration. While the method section mentioned that the longest one has more than 15 years of experience, the readers may be concerned about the other two raters who may have less experience. This information is crucial for assessing the credibility of this study.

Minor Concerns:

  1. The analysis performed in this study is comprehensive; however, the tables contain too many numbers and are difficult to read. Is it possible to simplify the results and focus on the major findings?
  2. The presentation of numbers in Tables should be uniform for better understanding. For example, in Table 1, the numbers were expressed as Number/%, but in Tables 2, 3, 4, and 5, it became %(number).
  3. Line 104 contains a typographical error: "with all three diameters @ 10mm..." I believe it should be "<" 10mm.

Author Response

Major Concerns:

  1. Please provide the general characteristics of the study population, including sex, age, mean thyroid nodule size, the exact number of patients in each TIRADS group, and the specific Bethesda classification of the nodules, not just the malignant FNA outcome (III-IV). This will allow the reader to have an overview of the enrolled patients.

Table 1 in the original manuscript contains information on patients’ sex and age, mean volume of their thyroid nodules and the number of nodules in each EU-TIRADS category. In the revised version Table 1 was extended to include information on the results of FNA of nodules from each EU-TIRADS category.

  1. In Table 3, the number of examined images (single transverse vs. both sections) did not affect the concordant rate of TIRADS categorization. Moreover, in rater A and C, the concordant rate is even higher in the single image group than the two-image group. Is there any possible explanation for this result? Theoretically, two images should contain more information and have a higher concordant rate compared to a single image.

Obviously, we agree that two static images bear more information than a single image. And it is logical to expect that two images should improve the accuracy of the assessment. However, that effect when compared to hundreds of images displayed subsequently on real-time imaging may be negligible. As our results indicate, the evaluation of two images does not lead to improved concordance of nodule classification on static US in relation to real-time US. That concordance was unsatisfactory and did not change significantly when two images were evaluated in relation to single image evaluation for any of the raters. The noted differences should be regarded as random. In our opinion, a significant improvement in the concordance of nodule categorization could be achieved only with the evaluation of stored video clips. That is way we wrote in the conclusions:

“Better inter-observer and inter-exam agreement may be obtained with the use of stored video clips that show whole volume of the nodule but this should be confirmed in future studies.”

Unfortunately, in many papers, information on what exactly was analyzed on static US is unavailable. And the results of our study indicate that this is a significant oversight.

Another question is the influence of the number of evaluated images on the inter-rater concordance. Intuitively, one may believe that more information should translate into better agreement of opinion between raters. But it is quite possible that the actual effect is opposite: the more information is analyzed, the higher the chance of inconsistent interpretations.

  1. Following the above question, is it possible that the study only enrolled patients who have two static images (n=545)? Simplifying the study by focusing on patients with two static images might make it easier to understand. Additionally, the concordant rate of assessing TIRADS in patients with different numbers of static images (1 vs. 2) could still be evaluated by giving the raters only one static image (transverse section) first, obtaining the first TIRADS classification, and then giving them the second still image (longitudinal section) for re-evaluation. If this cannot be done, please address this point in the limitations section.

The approach proposed by the Reviewer is not consistent with the purpose of our study, which was to answer the question whether the distribution of nodules into particular categories of EU-TIRADS is the same regardless of real-time US or static US evaluation and whether there is a difference between one and two images being analyzed on static US.

The study design proposed by the Reviewer actually answers another research question – whether the assessment of two images increases the frequency of categorization of nodules into categories with a higher risk compared to the assessment of one image. The answer to this question intuitively seemed obvious to us. It is difficult to assume that the reclassification of the nodule after viewing its image on the second cross-section (in this case longitudinal) will negate the presence of high-risk features previously recognized on the cross-section. The absence of such a feature on the longitudinal section could hardly be a basis for negating its presence on the transverse section. Therefore, with the proposed study design we would create the possibility of increasing the category of the nodule, and never or almost never to lower it. With such a model, we could show that the interpretation of two images more often gives a higher category EU-TIRADS, in particular allows for more frequent recognition of category 5 than the interpretation of one image. Actually, we observed such a regularity and inspired by the Reviewer’s comment we added the following sentence to the revision:

“But in the case of the most suspected nodules (classified into category 5 on rtUS) we observed that the fully concordant classification by three raters was slightly more often when they analyzed two images than only one image: 50.6% (40 out of 79) vs. 36.1% (13 out of 36), respectively (p=0.1474). In the case of nodules classified into category 3 or 4 on rtUS the observed differences were smaller and the effect of the number of analyzed static images was reverse.”

just after the sentence: “We didn’t find any significant difference in the rate of categorization agreement in relation to nodule’s size, nor the number of examined images (Table 5)” – line 238 of the original.

That issue was additionally mentioned in the Discussion in the sentence inserted after line 283 of the original.

We also moved the original Table 5 to supplementary materials (as Table S3), following the suggestion of another reviewer.

  1. Please specify the three raters' experience in performing ultrasound and fine needle aspiration. While the method section mentioned that the longest one has more than 15 years of experience, the readers may be concerned about the other two raters who may have less experience. This information is crucial for assessing the credibility of this study.

Indeed, the sentence describing the experience of the raters was not clearly formulated and could be misleading. All three rates had at least 15 years’ experience  in performing US imaging and FNA of the thyroid. We amended that sentence accordingly – at lines 87-90 of the original.

Minor Concerns:

  1. The analysis performed in this study is comprehensive; however, the tables contain too many numbers and are difficult to read. Is it possible to simplify the results and focus on the major findings?

Our analyses yielded a lot of data, of which we showed only a part. We are aware that despite this selection, some tables still contain a lot of numbers, but this is because we examined many ultrasound risk features separately and had to calculate particular Krippendorff's alpha coefficients and percentages of concordance for all of them. We would like to maintain this approach, it was appreciated by one of the reviewers.

  1. The presentation of numbers in Tables should be uniform for better understanding. For example, in Table 1, the numbers were expressed as Number/%, but in Tables 2, 3, 4, and 5, it became %(number).

Table 1 shows demographic characteristics of examined patients and the numbers of nodules classified into each category. That is why we intentionally presented that information as Number/% because absolute numbers are more relevant for such data. In the case of other tables percentages are more important and natural for the interpretation of presented data because these tables show concordance of nodule classification with various methods or different raters. That is why we opt to stay with the original version of these tables, but we added percent sign to the corresponding numbers in Table 1 for better clarity.

  1. Line 104 contains a typographical error: "with all three diameters @ 10mm..." I believe it should be "<" 10mm.

The indicated error occurred when the manuscript was pasted into the template prepared by the editors. We apologize for overlooking it, the error has been corrected.

Round 2

Reviewer 2 Report

The reproducibility is clinically meaningless if the impact on the FNA recommendation is not known. It's unfortunate that with all the data available to the authors, the authors have declined to analyze further on how the static image review differs in FNA recommendation. I strongly disagree with the authors’ response saying that such analysis is only meaningful with the postoperative histopathology results. Though the FNA Bethesda category is not the final diagnosis of thyroid nodules, it is a final report to determine how the thyroid nodules should be managed in clinical practice. If the authors are determined that their current analysis is sufficient, I do not believe the manuscript in its current form is acceptable to publish. 

Reviewer 3 Report

 Accept in present form